# Machine Learning-Enabled Prediction of 3D-Printed Microneedle Features

**DOI:** 10.3390/bios12070491

**Published:** 2022-07-06

**Authors:** Misagh Rezapour Sarabi, M. Munzer Alseed, Ahmet Agah Karagoz, Savas Tasoglu

**Affiliations:** 1Graduate School of Sciences & Engineering, Koç University, Istanbul 34450, Turkey; msarabi19@ku.edu.tr (M.R.S.); akaragoz22@ku.edu.tr (A.A.K.); 2Boğaziçi Institute of Biomedical Engineering, Boğaziçi University, Istanbul 34684, Turkey; munzer.alseed@boun.edu.tr; 3Koç University Translational Medicine Research Center, Koç University, Istanbul 34450, Turkey; 4Koç University Arçelik Research Center for Creative Industries, Koç University, Istanbul 34450, Turkey; 5Physical Intelligence Department, Max Planck Institute for Intelligent Systems, 70569 Stuttgart, Germany

**Keywords:** microneedles, machine learning, deep learning, 3D printing, artificial intelligence, image processing

## Abstract

Microneedles (MNs) introduced a novel injection alternative to conventional needles, offering a decreased administration pain and phobia along with more efficient transdermal and intradermal drug delivery/sample collecting. 3D printing methods have emerged in the field of MNs for their time- and cost-efficient manufacturing. Tuning 3D printing parameters with artificial intelligence (AI), including machine learning (ML) and deep learning (DL), is an emerging multidisciplinary field for optimization of manufacturing biomedical devices. Herein, we presented an AI framework to assess and predict 3D-printed MN features. Biodegradable MNs were fabricated using fused deposition modeling (FDM) 3D printing technology followed by chemical etching to enhance their geometrical precision. DL was used for quality control and anomaly detection in the fabricated MNAs. Ten different MN designs and various etching exposure doses were used create a data library to train ML models for extraction of similarity metrics in order to predict new fabrication outcomes when the mentioned parameters were adjusted. The integration of AI-enabled prediction with 3D printed MNs will facilitate the development of new healthcare systems and advancement of MNs’ biomedical applications.

## 1. Introduction

Microneedles (MNs), which are basically needles in the scale of micrometers at most, are robustly growing [1] pioneers of the minimally invasive injection procedures, enabling more advanced delivery/collecting systems [2,3]. In comparison with conventional needle technologies, MNs are verified to be more advantageous by decreasing the administration pain [4] and curtailment of anxiety and needle phobia [5], along with being more cost-efficient for drug delivery [6]. Furthermore, transdermal and intradermal injections using MNs are more efficient since they can directly access the skin dermis by penetration through the stratum corneum [7]. Among the various techniques developed for microneedle arrays (MNAs) fabrication, 3D printing is an emerging accessible method for their time- and cost-efficient manufacturing [8,9]. Available in different technologies such as fused deposition modeling (FDM), stereolithography (SLA), digital light processing (DLP), and two/multi-photon polymerization (TPP/MPP), 3D printing can be used for the direct fabrication of MNAs using a variety of printable materials [10,11,12,13] such as hydrogels, acrylates, epoxides, polyamides, polyurethane (PU), polylactic acid (PLA), acrylonitrile butadiene styrene (ABS), and also bioinks with the ability to encapsulate cells.

Artificial intelligence (AI), in which the machines are taught to simulate human intelligence and propose solutions to an issue or predict the outcome of a process by analyzing a series of input data, is gradually influencing healthcare systems and biomedical applications [14,15,16,17,18], such as disease prediction [19,20], diagnosing cancer [21,22], and medical imaging using AI [23]. Machine learning (ML) is a branch of AI which is utilized to detect patterns and relations in data using algorithms intended to make accurate predictions [24]. Deep learning (DL) is also part of the ML family based on artificial neural networks with representation learning. ML techniques are available in four major categories: supervised, semi-supervised, unsupervised, and reinforced [25]. ML is a prominent tool to understand the effect of different 3D printing parameters on the final product [26]. For example, ML was used for 3D-printability analysis in order to detect possible errors in the design [27], tuning microstructure to obtain desired mechanical properties [28], optimizing the energy expenditure in the 3D printing process [29], designing 3D printed surrogates that imitate the implementation of a target setup [30], maximizing adhesion of 3D printed biomimetic microfibrillar adhesives by optimizing their designs [31], and prediction of drug-releasing process from 3D printed medicines [32]. In this regard, AI techniques can also be trained with the geometrical details of MNs and their 3D printing process parameters for predicting and tuning the desired product; a realm not explored by the research teams so far.

In this study, we fabricated biodegradable PLA MNs utilizing FDM 3D printing (Figure 1A,B). The 3D-printed MNAs were then etched using potassium hydroxide (KOH) solution to enhance their geometrical features (Figure 1C,D). By causing variations in geometrical features (base diameter, height and drafting angle of MNs) and etching dose (etching solution concentration and etching duration), the studied cases included 240 arrays with 2400 MNs. To the best of authors knowledge, this is the first study presenting implementation of ML into the optimization of MN fabrication. In this regard, the dataset of the 2400 MNs was utilized for optimization and prediction of 3D-printed MNAs using AI methods. DL was trained with this dataset for quality control and anomaly detection in the fabricated MNAs. Next, using ML, the data was processed to extract similarity metrics between the designed MNAs and fabricated MNAs. The roles of geometrical specifications, and etching parameters were investigated using ML to predict the outcome of new prints when the mentioned criteria were altered (Figure 1E). Finally, the capability of the fabricated PLA MNAs’ insertion into the skin along with drug delivery capability were investigated. Integration of the two emerging sciences of MNs with AI will pave the way for easier development of advanced healthcare setups.

## 2. Methods

### 2.1. Designing and 3D Printing of the MNAs

The MNAs were designed with Dassault Systèmes (3DS) SolidWorks computer-aided design (CAD) and computer-aided engineering (CAE) software. The shape of the MNAs included ten different designs, by varying the parameters of: the needle base diameter, the needle height, and the angle of the draft (Table 1). The arrays were designed with one row comprised of 10 MNs. The MNAs were fabricated with FDM technology by Zaxe Z1 3D Printer (Zaxe 3D Printing Technologies, Istanbul, Türkiye) using biodegradable PLA filament purchased from the same company. The layer height of the 3D printer was set to 0.06 mm. The filling density parameter of the 3D printer was not applicable in the prints as the MNAs are smaller than to be filled (e.g., they are composed of only layer walls).

### 2.2. Preparing KOH Solution and Performing Etching Experiment

KOH in the form of flakes with 90% purity (Sabunsu, Antalya, Türkiye) was added to distilled water to result the KOH solution. Four solutions with various concentrations (3, 4, 5, and 6 M) were prepared. The 3D printed MNs (made from PLA) were completely immersed in the prepared KOH solutions for six various durations (4, 7, 14, 18, 21, and 24 h) followed by several washes with water to remove the KOH solution.

### 2.3. Image-Based Classification and Defect Detection Using DL

Photo of the each MNA was captured with a digital camera, resulting in 240 images of the fabricated MNAs (Figure 2A) to be fed to the DL. Python programming language was used in the DL technique. Each MNA, which was composed of ten needles, was divided into single MN photos with size of 150 × 150 pixel, using an image-processing pipeline, creating a dataset with 2400 instances. Each instance of this dataset was labeled as “non-defective” or “defective” by expert review, resulting in 1004 non-defective and 1396 defective images, which were all then binarized and split for training (75%), validation (10%) and testing (15%). Three different methods, classical ML, DL, and, unsupervised anomaly detection, were implemented for classification. For classical ML, stochastic gradient descent (SGD), decision trees, naïve Bayes, multi-layer perceptron (MLP), and support vector machines (SVM) (Table 2) were used upon flattening 150 × 150 binarized image dataset to 1 × 22,500 feature vector followed by a principal component analysis (PCA) method for reducing the feature dimensions from 22,500 to 10 to avoid the curse of dimensionality. Each algorithm was run several times by randomly splitting the dataset and averaging the output scores.

Utilizing Python, three different image classification networks based on Convolutional Neural Networks (CNNs), Resnet34 [33] as the baseline, MobilnetV2 [34] as the light network, and ConvNeXt_Base [35] as state-of-the-art network were used in this work (Table 3). These networks were implemented through PyTorch [36] with transfer learning method by using pre-trained networks on ImageNet [37]. Since ImageNet has 1000 classes, those networks’ last layer contains 1000 dimensions. Therefore, we replaced every model’s last fully connected layer to our sequential model, which composed of a 1024 × 128 linear layer, a dropout layer with 0.4 probability, followed by a 128 × 2 linear layer. SGD optimizer and binary cross entropy loss was used with 24 batch size and 0.001 learning rate to fine-tune the network on 20 batches. For the task of anomaly detection, Patch Distribution Modeling (PaDiM) algorithm [38] was applied (Figure 2B). PaDiM utilizes the pretrained CNN for patch embedding, and of multivariate Gaussian distributions to get a probabilistic rendering of the normal class. PaDiM uses only non-defective samples for testing in order to learn the distribution of normal images, followed by assigning anomaly scores to the images in the testing period. Only 450 non-defective MN images were used in training and the rest were used for testing.

### 2.4. Extracting Similarity Indices Using Image Processing

For image processing, a total of 240 instances were used, represented by the different criteria of Table 1. Images of the fabricated MNAs were named as “3DP images”. Additionally, to compare the original designs with the fabricated MNAs, side shot of each CAD was taken, resulting in 240 “CAD images”. Each 3DP image was manually aligned to its corresponding CAD image using Adobe Photoshop, resulting in 240 3DP-CAD pairs. MathWorks MATLAB was utilized for processing each pair in the next step (Figure 2C). A MATLAB code was developed to convert the images into binary and compare their pixels, giving a set of true positives (TPs), true negatives (TNs), false positives (FPs) and false negatives (FNs). A pixel was considered to be “positive” if it belonged to a needle, otherwise it was considered to be “negative”. On the other hand, pixels were considered as “true” if they had the same binary values in both images, otherwise they were considered as “false”.

According to the confusion matrix of each image pair, five sets of metrics were calculated (Table 4). The first metric (Similarity Index) measured the similarity error, calculated by dividing the sum of false pixels (FP and FN) by the area of the MN pixels in the CAD image, then subtracting that error from one. The second metric (3DP-to-CAD Area Ratio) calculated the ratio between the areas of MN pixels of the CAD image and 3DP image. Accuracy as the third metric measured the ratio between true pixels (both positive and negative) and all of the pixels in one image. The fourth metric (Sensitivity) calculated the ratio between the pixels that only represented MNs in both images (i.e., TP) and the area of the MN pixels in the CAD image. Finally, the Dice Similarity Coefficient was utilized to measure the ratio between intersection and union of the two images. i.e., the true positives over the MN pixels in both images.

### 2.5. Creating Training Library and Applying ML

MATLAB’s Regression Learner and Regression Learner toolboxes were used for choosing and training the ML models. For the regression task, five folds were used for cross validation, where the data set was divided into five subsets, and the ML algorithm was trained five times. At each training round, one of the five subsets was considered as the validation set, while the rest data was used for training, resulting in five R^2^ values, of which the mean was calculated as the reported R^2^ value. All five features were used without any dimensionality reduction, which included three geometric features (height, base diameter and drafting angle of microneedle) and two etching features (solution concentration and etching duration).

A separate training session was applied for each single similarity metric by applying all available models in the toolbox including linear regression, neural networks, regression trees, support vectors machines, and Gaussian process regression, and choosing the one that resulted in the highest R^2^ value. For classification, the sensitivity metric which ranges between 0 and 1 was chosen as the similarity index, and a threshold of 0.8 was set to convert the data into binary, where similarity indices below 0.8 were considered to be low (Class 0), otherwise they were considered as accepted similarities (Class 1). All available models in the MATLAB’s Classification Learner toolbox were used, and five folds were used for cross validation. Since classification trees were found to yield the highest accuracies, a classification tree was constructed to optimize its hyperparameters, which are the maximum number of splits and split criterion, and then it was trained for classification (Figure 2D).

### 2.6. Skin Perforation Experiment on Porcine Skin Ex Vivo

Fresh porcine flank skin pieces were supplied from a local butcher’s shop and were transferred with ice to the laboratory and were kept frozen in −20 °C refrigerator. On the experiment day, the skin was thawed at room temperature. Upon after, the skin was cut into pieces with appropriate sizes to be put on the device. The insertion of MNAs into the skin was performed using Instron ElectroPuls Model E10000 Axial Torsion Test System (Illinois Tool Works Inc., Glenview, IL, USA) in compression mode with precise measurement of the required force for the skin perforation. Custom holders were fabricated using 3D printing for placing the skin samples onto the shafts of the device. Data was collected every 20 ms. Furthermore, in order to keep the skin fresh and prevent its dehydration during experiment, skin samples were kept on paper tissues soaked with liquid Dulbecco’s phosphate-buffered saline w/o calcium w/o magnesium (Biowest, Lakewood Ranch, FL, USA).

### 2.7. Preparing Rhodamine B (RhB) Solution and Its Delivery as a Drug Model

Food wheat starch (Tibet İthalat İhracat ve Kozmetik, Istanbul, Türkiye) in two values of 30 g and 15 g were added to two beakers with 75 mL distilled water. Both mixtures were put on a magnetic hot plate stirrer (Hangzhou Miu Instruments, Hangzhou, China) at 100 °C and 400 rpm stirring, for 24 min. The solution with 15 g starch was chosen as the model drug carrier. 0.075 g RhB (REF 1.07599.0025, Sigma Aldrich, St. Louis, MO, USA) as the model drug was added to the starch solution and was completely stirred to result a uniform solution. Tips of an example MNA (case 6 in Table 1, featuring a designed based diameter and height of 1500 μm and 2000 μm, respectively, and draft angle of 5°, etched in a 5 M KOH solution for 14 h) were coated with the prepared solution and were inserted into the porcine skin with hand. In order to keep the skin fresh and prevent its dehydration during experiment, skin sample was kept on paper tissues soaked with liquid Dulbecco’s phosphate-buffered saline w/o calcium w/o magnesium (Biowest, Lakewood Ranch, FL, USA). After removing the MNA from the skin, the perforation site was imaged under visible light, and under ultraviolet (UV) light (OmniCure S2000, Excelitas Technologies, Waltham, MA, USA).

## 3. Results and Discussion

### 3.1. Role of Design Parameters and Etching Doses

The MNAs were designed in ten different shapes with altering the needle base diameter, the needle height, and the angle of the draft (Table 1, Figure 1A). MNAs were 3D-printed with PLA, a biodegradable thermoplastic polyester. The utilized 3D printer for the fabrication process, was a desktop FDM model (Figure 1B). While being cost-efficient, accessible, and easy-to-use, the 3D printer has some resolution limitations in printing tiny features such as MNs. The device showed better capability in printing cylindrical shaped MNs (with a draft angle of zero) in comparison with conical shaped MNs (with the designed draft angle of 5° or 10°). However, in all cases, the nozzle of the printer left material traces between the needles which created some imperfections in the arrays (Figure 1C(i)). In order to eliminate these imperfections, KOH solutions with various concentrations (3, 4, 5, and 6 M) and various exposure durations (4, 7, 14, 18, 21, and 24 h) were used for etching the 3D-printed PLA MNs (Figure 1C(ii)). The etching process resulted in elimination of the mentioned imperfections and enhanced the geometrical features (Figure 1C(iii)), in accordance with previous reports of fabrication of MNs using 3D printing followed by etching to reach ideal size/shape [39,40].

In addition, as the concentration of the KOH solution was increased, the etching process demonstrated better enhancement of the MNA features. Also, exposure duration exhibited a similar result, in which longer duration resulted in more removal of the flaws. However, the strongest KOH exposure (e.g., 6 M concentration for 24 h duration) resulted in making the high-aspect ratio MNAs (e.g., cases 9 and 10 in Table 1) more brittle, as the process removed some of the needle bodies as well. In general, mild conditions of etching exposure were helpful to achieve more perfect products similar to the initial CAD design. For example, the MNA featuring a designed based diameter and height of 1000 μm and 2500 μm, respectively, and draft angle of 5° (case 4 in Table 1), etched in a 5 M KOH solution for 18 h, resulted in the measured average base diameter and height of ~952 μm and ~2494 μm, respectively (Figure 1D). In total, by variation in geometrical features and etching dose, the studied cases included 240 arrays with 2400 MNs, in which they were analyzed using DL for detection of fabrication defects followed by training with ML in order to create a model to predict the outcome of new prints (Figure 1E).

### 3.2. Anomaly Detection Using DL

Defect detection plays an important role in autonomous production systems. DL models were utilized here to identify defective MNAs among 2400 fabricated MNs. The main task of anomaly detection algorithms is to identify abnormal images and localize abnormal areas. Most industries prefer unsupervised anomaly detection methods over supervised ones since not only collecting and labelling anomaly data is very expensive and time-consuming, but also supervised methods lack generalization capabilities. First, classical ML was used with methods including SGD, Decision Trees, Naïve Bayes, MLP, and SVM (Table 2). After running the algorithms using python programming and open-source libraries such as Scikit-learn [41], PyTorch [36], and NumPy [42], the results were evaluated based on metrics such as precision, recall, F-1 score, and accuracy. The results indicated that the best performed algorithm was SVM with 0.86 accuracy followed by MLP and Naïve Bayes.

Upon fine-tuning pre-trained CNNs at approximately 15–20 epochs, the test set was classified by trained candidate CNNs (e.g., Resnet34, MobilnetV2, and ConvNeXt_Base), and the same performance metrics as ML methods were evaluated (Table 3). By comparison of results at Table 2 and Table 3, it can be demonstrated that the CNNs outperformed ML methods. In addition, it can be observed that the best performance was for the state-of-the-art ConvNeXt_Base model with 0.96 accuracy, since it contains a complex deep network. Resnet34, considered as the core standard backbone classification network, also performed close to ConvNeXt_Base with 0.92 accuracy. On the other hand, MobilnetV2, which is a tiny network suitable for mobile devices, got slightly higher scores than Resnet34 with ten times less feature size. Despite the deep CNNs considered as black boxes, activations on layers are presented in Figure 3A,B, with activation maps visualized with the overall defect detection pipeline.

For anomaly detection, only 450 random images selected from non-defective images were used in training, while the others were reserved for testing. PaDiM used a pre-trained ResNet backbone to extract the features from images and learnt the distribution of non-defective images with a multivariate gaussian type of algorithm. Therefore, there was no training loop for this method, meaning that non-defective images were fed into the algorithm and the algorithm calculated the parameters at once. The Receiver Operating Characteristics-Area Under The Curve (ROC-AUC) score is a performance measurement for the classification problems. ROC-AUC score was used to evaluate the anomaly detection task, as it represents the PaDiM algorithm’s classification capability better than other metrics. Upon testing, 0.919 image level ROC-AUC score have achieved by PaDiM algorithm. As PaDiM generates an anomaly score for each pixel, the anomaly maps pointing out abnormal areas defined by the algorithm were visualized (Figure 3C).

### 3.3. Image Processing and ML

For image processing, a total of 240 instances were used, represented by the different fabrication criteria mentioned before. Five similarity metrics (similarity index, 3DP-to-CAD area ratio, accuracy, sensitivity, and Dice similarity coefficient) were calculated from the results of image processing (Table 4). Among these metrics, 3DP-to-CAD area ratio yielded the highest R^2^ value. This can be a result of representing only the increase or decrease of the 3D-printed MN in comparison with CAD due to overprinting or etching, respectively. Also, this metric does not have a specific upper bound and can exceed 1 in many cases. On the other hand, the other four metrics yielded R^2^ values of no more than 0.75 with a minimum of 0.55, which do not indicate a good fitting of the models to the available data. The reason for having poor R^2^ values can be explained by the fact that the dataset is relatively small. In addition, a considerable amount of noise seems to affect the data, initiating from different causes, such as the bridging between MNs during 3D printing, the heterogeneity of etched regions, and the damage caused to some MNs randomly during etching.

Sensitivity metric was therefore selected as the similarity index since it yielded the second best R^2^. However, its R^2^ of 0.75 was not high enough for an acceptable regression, and hence the data was converted to binary by setting a threshold of 0.8 to split the data into two classes. Optimizing the hyperparameters of a classification tree with the existing binary data resulted in a maximum number of splits of 20, while Gini’s diversity index was found to be the optimal split criterion. The confusion matrix generated after applying classification (Table 5), indicated that 128 out of the 135 images with high quality and 98 out of the 105 images with low quality were classified correctly, leaving 14 misclassified images out of the whole images (seven false negatives and seven false positives). The confusion matrix also showed that classification improved the prediction of the ML model, with an accuracy of 94.17%, sensitivity of 94.81%, and specificity of 93.33%. However, the drawback of using classification is that the exact similarity between the designed and fabricated MN is not reported, but only the quality of fabrication is reported as high or low.

Figure 4 represents the results of processing the images. The colors illustrate the difference between true positives, true negatives, false positives, and false negatives (Figure 4A). Example results of image processing for one of the studied designs (Case 8 from Table 1) is shown in Figure 4B–E. It can be demonstrated that increasing the etching solution concentration or increasing the etching duration have similar effects of decreasing the false positives (i.e., excess printed material) and increasing the false negatives (i.e., missing printed material). To allow the final trained classification ML model to be efficiently used by users to predict 3D-printed MNAs quality, a graphical user interface (GUI) was designed using MATLAB’s App Designer tool (Figure 5A). GUI aimed to enable users with no programming experience to utilize the ML model easily without having to write codes. The GUI included an illustration of an example MNA to explain the meaning of each of the five input parameters to the user. A text box with the name of each of these parameters was introduced, where users can enter the values at will. The trained and optimized classification tree model was integrated with the GUI to perform the prediction. After receiving the five input parameters, and upon the user clicking on the button “MNA Quality”, the parameters are processed with the classification tree, calculating the predicted MNA quality and finally showing the predicted result next to the button as “High” or “Low”. In addition, upper and lower ranges for each of the input parameters were introduced to the GUI. For example, etching solution concentration was limited to the range of 3 M to 21.6 M, as concentrations lower than 3 M are not so efficient in etching, and 21.6 M is the maximum concentration of KOH in water that can be prepared in 25 °C. Therefore, GUI would reject out-of-range values for the parameters.

### 3.4. Capability of MNAs for Perforation of Porcine Skin Ex Vivo

Human skin is composed of three layers. The outermost layer which is called the epidermis, a waterproof barrier and representative of the skin tone. The dermis, the second layer located under the epidermis, contains sweat glands, arrector pili muscles, hair follicles, and sebaceous glands [9]. The third layer which is called the hypodermis is the deeper subcutaneous tissue consisting of fat and connective tissues [9]. Since using human skin in experiments faces ethical issues and requires laboratory approval, polymeric films, paraffin wax, agarose gel, rat skin, and porcine skin have been reported in literature to be widely utilized instead for experiments simulating the human skin [43]. Porcine skin is histologically similar to the human skin, and the thickness of porcine flank skin epidermis, 68 ± 26 µm, is also similar to the human skin epidermis thicknesses, 68 ± 26 µm in face, 65 ± 24 µm in neck, and 68 ± 21 µm in arms [44]. Hence, it can be used as an alternative for human skin with an appropriate resemblance. In this regard, porcine flank skin was used in this study, and the capability of the fabricated PLA MNAs’ insertion into the skin was investigated.

All ten alternative designs of Table 1, etched in 6 M KOH solution for 14 h, were tested for their skin penetration capability. Custom 3D-printed holders were fabricated for placing the skin samples and the MNAs on the device (Figure 5B). The MNA was bonded to the upper holder, and the porcine skin sample was positioned under the MNA on the lower holder (Figure 5C). The device moved these holders towards each other gently until the MNs were completely inside the skin. The required amount of force for perforation of the skin was measured precisely (Figure 5D). Among the designs with difference only in their drafting angle (e.g., cases 1 and 2, and cases 5 and 6), the results indicated that less force was required for insertion into the skin as the drafting angle increased. On the other hand, cases 3 and 7, which had zero drafting angle, were not able to perforate the skin sample and broke under the increasing force. The results can be interpreted from the point view of ratio of the needle height to the needle base diameter as well, an index similar to aspect ratio. As this ratio increased, less amount of force was required for penetration. For example, cases 1, 2, and 10 (with a ratio equal to 2) needed more effort for skin perforation in comparison with cases 4 and 9 (with ratios of 2.5 and 3, respectively). Furthermore, maximal static pushing force for humans was reported to be in the 583–1285 N range with both hands, and 262–520 N with their preferred hand [45]. Hence, all of the MNAs discussed here, can be easily inserted by human hand as well.

### 3.5. Capability of MNAs for Model Drug Delivery into Porcine Skin Ex Vivo

In order to investigate the capability of the fabricated MNAs for delivery of a drug candidate into skin, tips of an example MNA (case 6 in Table 1, featuring a designed based diameter and height of 1500 μm and 2000 μm, respectively, and draft angle of 5°, etched in a 5 M KOH solution for 14 h) were coated with an aqueous solution of starch and RhB (Figure 5E). Previous reports have stated using thickener agents to facilitate deposition of the model drug onto the surface of the needle tips [46,47]. Starch was chosen for this purpose in the current study, as it is highly available, cost-efficient, and a biodegradable option well-known in MN studies [48]. Starch solution was prepared with two amounts of 30 g and 15 g. The solution containing higher value of starch, behaved as a viscoelastic solid (e.g., highly viscous). Hence, lower concentration of starch solution was chosen as it showed an appropriate viscosity for deposition on MNAs. The coated MNA was inserted into porcine skin ex vivo, with hand. Upon removing the MNA from the skin, the MNA perforation site was imaged under visible light (Figure 5F), and under UV light (Figure 5G). It can be observed that the RhB as a model drug was successfully delivered into the skin.

## 4. Outlook and Conclusions

DL and ML techniques with the capability of identifying patterns in the input information, administering multi-dimensional and multi-variety data, and predicting new outcomes coupled with incessant improvement are shaping the future of data analysis approaches. In order to enable users with no programming background to practically benefit from the developed codes using ML and DL techniques, translation of these codes into proper computer/mobile applications is the next step. This can empower the trained models to be more applicable for future research purposes. For example, developed DL-based quantification of clay minerals was integrated into an application for a better user experience [49]. For the case of MNAs, ML models can predict the quality of the needles even before designing, resulting in cost- and time- efficient procedures with observant usage of materials and resources. In this study, after we fabricated PLA MNs with FDM 3D printing, followed by chemical etching, we were able to achieve predictability in the design and fabrication procedure using ML models. It can be easily concluded that by using ML models, producing perfect MNAs can be planned in a faster way, which will eventually result in faster adaption with medical applications. We achieved the predictability code using five set of parameters here (base diameter, height and drafting angle of MNs, and etching solution concentration and etching duration), which was the base for the GUI for user-friendly implementation of the code. The investigated required force for penetration of MNAs into skin can be modeled using ML algorithms as well in order to predict the MNAs’ capability for insertion into the skin, as a guide for future studies. In a nutshell, this study can come to an end stating that integration of AI science with MNs will pave the way for easier development of advanced healthcare systems.

## Figures and Tables

**Figure 1 biosensors-12-00491-f001:**
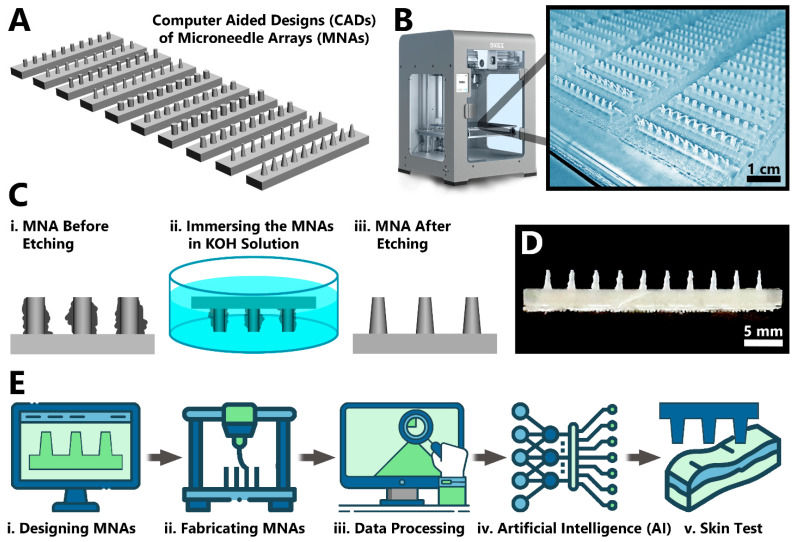
Fabrication of MNAs using 3D printing and features enhancement using chemical etching. (**A**) Ten different geometries of MNs were drawn using CAD, featuring various needle base diameters, needle heights, and draft angle (Table 1). (**B**) The MNAs were fabricated with FDM 3D printing technology using biodegradable PLA. (**C**) Since the printed needles had some imperfections (**i**), they were etched using KOH solution (**ii**), in order to remove the extra parts and thinner the needle bodies (**iii**). (**D**) An example final MNA featuring a designed based diameter and height of 1000 μm and 2500 μm, respectively, and draft angle of 5°, etched in a 5 M KOH solution for 18 h (case 4 in Table 1). The measured average base diameter and height of the needles for this case were ~952 μm and ~2494 μm, respectively. (**E**) The general workflow of the present study: After (**i**) designing the MNAs, and (**ii**) fabricating them using 3D printing followed by etching, (**iii**) their geometrical specifications and etching parameters processed using AI methods (**iv**) in order to predict the outcome of new prints, followed by (**v**) testing the fabricated MNAs for their skin perforation capability.

**Figure 2 biosensors-12-00491-f002:**
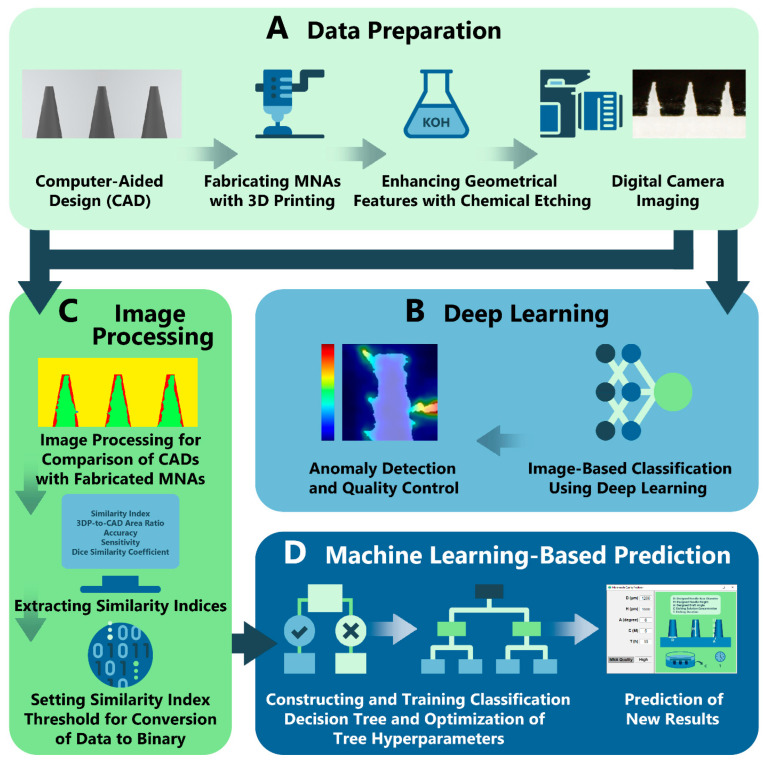
Workflow of preparing and processing data for AI models. (**A**) Based on design and fabrication criteria, a total of 240 MNA instances were possible. Each instance was imaged with a digital camera. (**B**) The captured MNA photos were automatically separated and binarized to a dataset of 2400 MN photos, which were analyzed for their quality and anomaly detection using DL. (**C**) For comparison of original designs and the fabricated MNAs, instances were processed for extraction of five different similarity metrics. ML models were developed by receiving five input parameters (three design features: base diameter, height and drafting angle of MNs; and two etching features: etching solution concentration and etching duration). (**D**) A classification tree was constructed to optimize the hyperparameters (maximum number of splits and split criterion), and then it was trained for classification. Finally, the trained model was used to predict the quality of the resulting MNA for the criteria defined by user.

**Figure 3 biosensors-12-00491-f003:**
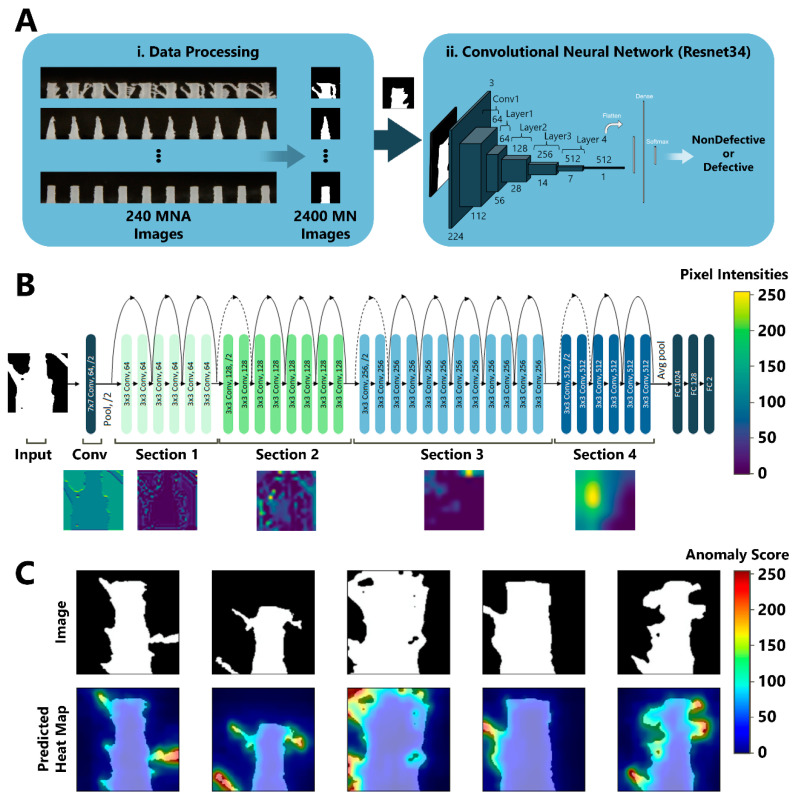
Image-based classification and anomaly detection using DL. (**A**) To use the MNA image dataset for the defect detection task, a data processing pipeline was implemented in which (**i**) each MNA image was automatically cut and binarized to separate individual MN images. Upon labeling all MNs as “Non-Defective” or “Defective” by expert review, the dataset was used for training CNN models. (**ii**) The resulting individual MN images were fed into Resnet34 CNN to be classified as “Non-Defective” or “Defective”. (**B**) Visualization of Resnet34 layer activation, where four ResNet sections are utilized in the CNN. The defective areas get higher pixel intensities throughout the network. (**C**) DL-based anomaly detection using PaDiM. Individual MNs were first binarized (**top row**) and then were used as an input for the PaDiM algorithm to generate heat maps of the anomaly scores of the MNs (**bottom row**). Anomaly scores are represented by pixel intensities between 0 (e.g., no anomaly) and 255 (e.g., extreme anomaly).

**Figure 4 biosensors-12-00491-f004:**
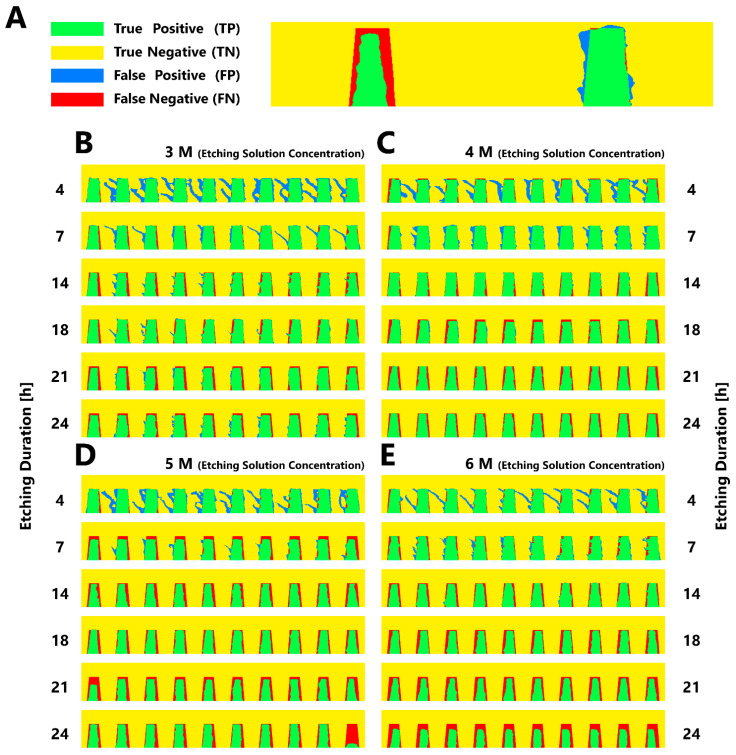
Image analysis procedure using MNA photos aligned with their corresponding CAD for purpose of extracting similarity indices. (**A**) Upon alignment of each experimental image to its corresponding CAD, a code was developed for extraction of several similarity metrics. The total instances which included 240 cases, were converted into binary and their pixels were compared, giving a set of TPs, TNs, FPs, and FNs. A pixel was considered to be “positive” if it belonged to a MN, otherwise it was considered to be “negative”. Also, pixels were considered as “true” if they had the same binary values in both images, otherwise they were considered as “false”. Example image analysis results for one of the studied designs (Case 8 from Table 1), exposed to KOH etching in four concentrations of (**B**) 3 M, (**C**) 4 M, (**D**) 5 M, and (**E**) 6 M, for six durations (4, 7, 14, 18, 21, and 24 h).

**Figure 5 biosensors-12-00491-f005:**
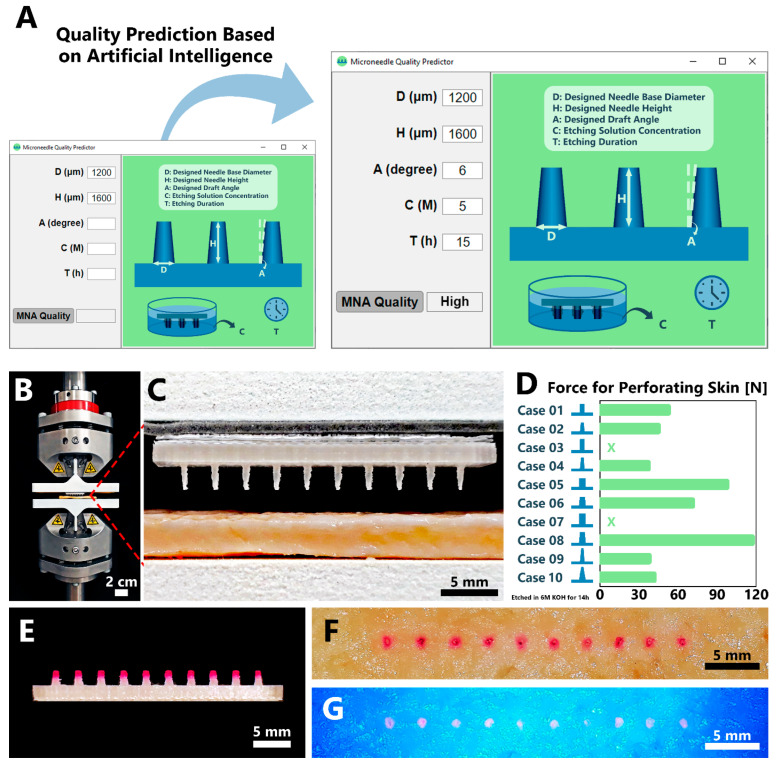
Developed GUI for MNA quality prediction, and capability of MNAs for perforation of porcine skin ex vivo. (**A**) A GUI was developed to predict the quality of the fabricated MNs based on the geometric and etching features input. The core of the GUI was based on the developed ML model, with addition of upper and lower limits for the input parameters (i.e., GUI rejects any input parameter that is out of its determined logical range). The user can enter the five parameters they intend to utilize for fabrications of MNAs, and upon clicking the button “MNA Quality”, the predicted quality of the MNA will be displayed next to the button as “High” or “Low”. (**B**) Using custom 3D-printed holders, (**C**) the skin sample was placed on the lower holder and the MNA was bonded to the upper holder. The device moved these holders towards each other until the MNs were inside the skin while (**D**) the required force [N] value for perforation of the skin was measured precisely for all of the geometrical specifications offered in Table 1. The MNAs used in this experiment were etched in 6 M KOH solution for 14 h. X: The MNA was not able to perforate the skin sample and broke under the increasing force. (**E**) In order to show the capability of the prepared MNAs for delivery of RhB as a model drug, tips of an example MNA (case 6 in Table 1, featuring a designed based diameter and height of 1500 μm and 2000 μm, respectively, and draft angle of 5°, etched in a 5 M KOH solution for 14 h) were coated with aqueous solution of starch and RhB mixture and were inserted into the porcine skin. The insertion and perforation sites are presented (**F**) under visible light, and (**G**) under UV light.

**Table 1 biosensors-12-00491-t001:** Geometrical parameters (needle base diameter, needle height, and draft angle) used to design various shapes of MNs, manufactured by 3D printing, followed by etching experiment with the presented criteria.

#	Schematics	Designed Needle Base Diameter (μm)	Designed Needle Height (μm)	Designed Draft Angle (deg)	Etching Solution Concentrations (M)	Etching Durations (h)
1	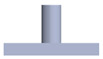	1000	2000	0	3; 4; 5; 6	4; 7; 14; 18; 21; 24
2	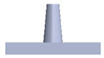	1000	2000	5	3; 4; 5; 6	4; 7; 14; 18; 21; 24
3	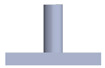	1000	2500	0	3; 4; 5; 6	4; 7; 14; 18; 21; 24
4	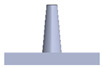	1000	2500	5	3; 4; 5; 6	4; 7; 14; 18; 21; 24
5	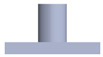	1500	2000	0	3; 4; 5; 6	4; 7; 14; 18; 21; 24
6	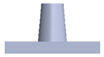	1500	2000	5	3; 4; 5; 6	4; 7; 14; 18; 21; 24
7	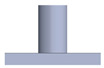	1500	2500	0	3; 4; 5; 6	4; 7; 14; 18; 21; 24
8	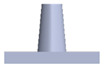	1500	2500	5	3; 4; 5; 6	4; 7; 14; 18; 21; 24
9	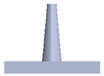	1000	3000	5	3; 4; 5; 6	4; 7; 14; 18; 21; 24
10	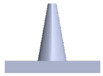	1500	3000	10	3; 4; 5; 6	4; 7; 14; 18; 21; 24

In total, the studied cases included 240 arrays with 2400 MNs.

**Table 2 biosensors-12-00491-t002:** Performance of ML algorithms for image classification task.

#	Model	Precision	Recall	F-1 Score	Accuracy
1	SGD	0.75	0.75	0.75	0.75
2	Decision Trees	0.76	0.76	0.76	0.76
3	Naïve Bayes	0.83	0.82	0.82	0.82
4	MLP	0.84	0.84	0.84	0.84
5	SVM	0.87	0.86	0.86	0.86

**Table 3 biosensors-12-00491-t003:** Performance of DL algorithms for image classification task.

#	Model	Precision	Recall	F-1 Score	Accuracy
1	Resnet34	0.93	0.92	0.92	0.92
2	MobilnetV2	0.94	0.93	0.93	0.93
3	ConvNeXt_Base	0.96	0.96	0.96	0.96

**Table 4 biosensors-12-00491-t004:** Performance of ML algorithms for the extracted similarity metrics.

#	Metric	Formula	Best Performing ML Model	R^2^
1	Similarity Index	$\frac{\mathrm{TP}-\mathrm{FP}}{\mathrm{TP}+\mathrm{FN}}$	SVM (Cubic Kernel)	0.63
2	3DP-to-CAD Area Ratio	$\frac{\mathrm{TP}+\mathrm{FP}}{\mathrm{TP}+\mathrm{FN}}$	Gaussian Processes (Squared Exponential)	0.90
3	Accuracy	$\frac{\mathrm{TP}+\mathrm{TN}}{\mathrm{TP}+\mathrm{TN}+\mathrm{FP}+\mathrm{FN}}$	SVM (Cubic Kernel)	0.60
4	Sensitivity	$\frac{\mathrm{TP}}{\mathrm{TP}+\mathrm{FN}}$	Gaussian Processes (Squared Exponential)	0.75
5	Dice Similarity Coefficient	$\frac{\mathrm{TP}}{\mathrm{TP}+\mathrm{FP}+\mathrm{FN}}$	Gaussian Processes (Squared Exponential)	0.55

**Table 5 biosensors-12-00491-t005:** Confusion matrix generated after binary classification.

		**Predicted Class**		
		0	1	
**True Class**	0	TN = 98	FP = 7	105
	1	FN = 7	TP = 128	135
		105	135	240

## Data Availability

The data that support the findings of this study are available from the corresponding author upon reasonable request.

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
