# Peer review of "Machine Learning-Enabled Prediction of 3D-Printed Microneedle Features"

_biosensors, 2022, doi:10.3390/bios12070491_

Round 1

Reviewer 1 Report

1. The English quality is good but minor amendments are required at times. 

2. The abbreviations should be interpreted in their first appearance.

3. The statement of the novelty and basis of the problem in the end paragraph of the introduction needs to be improved. In the current form, it is not clear how the present study adds to the existing knowledge.

4. Were the models trained using Python or MATLAB, it needs to be further clarified.

5. A very major problem with the application of machine learning techniques in the nowadays scientific context is that these models are only useful for the user himself and the practical application of the model in cases other than that of the conducted research is not considered. Therefore, a recent approach has been developing a user-friendly App for the developed model which prevents the complicated and tedious data preparation task for a new application of the model by the future user. The authors might enrich the text by mentioning this challenge, i.e. in order to make the trained models more applicable for future research purposes, the trained models could be translated into proper Apps for which a typical example could be found in https://doi.org/10.1016/j.energy.2021.122599 . Even though this comes from the field of microporous media, I do believe it could be adopted in all other fields as far as the machine learning approach is engaged. I understand the authors might not be able to provide such an App, but at least such a typical example needs to be introduced to the reader for future works. If available, the trained model of the present study would be provided as a supplementary file for the case that the reader aims to use it and does not need to do training from the scratch again.

6. Figure captions are too long

Author Response

We thank the reviewers for valuable suggestions to improve our manuscript. We addressed all the comments of the reviewers. Our point-by-point responses are listed inside the attached file.

Reviewer 2 Report

- moderate English amendments are needed throughout the manuscript

- 'DL' and 'ML' are acronymised a few times throughout the manuscript. This should be done at the first instance of their use.

- The methodology can be expanded across both physical and computational experiments.

- Figure 3Aii is this an original image? If not then it needs to be appropriately referenced

- Table 5 can be improved. It appears too condensed.

- Figure 5D  has no error bars

Author Response

(The authors gave the same response as above.)

Round 2

Reviewer 1 Report

Acceptable.